# The effect of different COVID-19 public health restrictions on mobility: A systematic review

**Mark A. Tully**[1]*, **Laura McMaw**[2], **Deepti Adlakha**[3], **Neale Blair**[4], **Jonny McAneney**[5], **Helen McAneney**[6], **Christina Carmichael**[7], **Conor Cunningham**[8], **Nicola C. Armstrong**[9], **Lee Smith**[7]

**1** Institute of Mental Health Sciences, School of Health Sciences, Ulster University, Newtownabbey, United Kingdom, **2** School of Health Sciences, Ulster University, Newtownabbey, United Kingdom, **3** Department of Landscape Architecture and Environmental Planning, Natural Learning Initiative, North Carolina State University, Raleigh, North Carolina, United States of America, **4** Built Environment Research Institute, Ulster University, Newtownabbey, United Kingdom, **5** Institute of Mental Health Sciences, Ulster University, Newtownabbey, United Kingdom, **6** UCD Centre for Interdisciplinary Research, Education, and Innovation in Health Systems, School of Nursing, Midwifery and Health Systems, University College Dublin, Belfield, Dublin, Ireland, **7** Centre for Health, Performance and Wellbeing, Anglia Ruskin University, Cambridge, United Kingdom, **8** Institute of Public Health, City Exchange, Belfast, United Kingdom, **9** Health and Social Care Research & Development Division, Public Health Agency (Northern Ireland), Belfast, United Kingdom

* m.tully@ulster.ac.uk

**Data Availability Statement:** Data sharing is not applicable to this article as no datasets were generated or analysed during the current study.

## Abstract

### Background

In response to the COVID-19 pandemic, most countries have introduced non-pharmaceutical interventions, such as stay-at-home orders, to reduce person-to-person contact and break trains of transmission. The aim of this systematic review was to assess the effect of different public health restrictions on mobility across different countries and cultures. The University of Bern COVID-19 Living Evidence database of COVID-19 and SARS-COV-2 publications was searched for retrospective or prospective studies evaluating the impact of COVID-19 public health restrictions on Google Mobility. Titles and abstracts were independently screened by two authors. Information from included studies was extracted by one researcher and double checked by another. Risk of bias of included articles was assessed using the Newcastle Ottowa Scale. Given the heterogeneous nature of the designs used, a narrative synthesis was undertaken. From the search, 1672 references were identified, of which 14 were included in the narrative synthesis. All studies reported data from the first wave of the pandemic, with Google Mobility Scores included from January to August 2020, with most studies analysing data during the first two months of the pandemic. Seven studies were assessed as having a moderate risk of bias and seven as a low risk of bias. Countries that introduced more stringent public health restrictions experienced greater reductions in mobility, through increased time at home and reductions in visits to shops, workplaces and use of public transport. Stay-at-home orders were the most effective of the individual strategies, whereas mask mandates had little effect of mobility.

### Conclusions

Public health restrictions, particularly stay-at-home orders have significantly impacted on transmission prevention behaviours. Further research is required to understand how to

The search strategy and search results have been provided as supplementary files.

**Funding:** The author(s) received no specific funding for this work.

**Competing interests:** The authors have declared that no competing interests exist.

effectively address pandemic fatigue and to support the safe return back to normal day-to-day behaviours.

## Introduction

On 30[th] January 2020, the Director-General of the World Health Organisation (WHO) declared the COVID-19 outbreak a public health emergency of international concern. COVID-19 is a respiratory disease caused by a virus known as SARS-CoV-2. This virus is characteristically highly contagious, with a doubling time of infected persons from the alpha variant of between six and seven days [1]. As of 15[th] June 2021, almost 176 million people have been infected worldwide and 3.7 million deaths have been reported [2].

To prevent the transmission of SARS-CoV-2 and reduce the subsequent morbidity and mortality from COVID-19, a range of measures have been introduced across the globe. In addition to testing and quarantining and as no vaccine was available, most countries have introduced a set of non-pharmaceutical interventions (NPIs) designed to reduce person-to-person contact and break trains of transmission through physical distancing. These included the introduction of stay-at-home orders, whereby citizens were instructed only to leave their residence for essential purposes such as grocery shopping or to seek medical care. Workplaces and non-essential businesses were closed, public gatherings prohibited, and restrictions were placed on non-essential travel [3].

Research shows that mobility and transport accessibility are highly correlated with social disadvantage [4]. During the pandemic, vulnerable populations (e.g., low-income households, disabled populations) and frontline, essential workers were more likely to use public transportation to travel to their place of work, increasing the risk of infection [5]. These issues are important because transport inequities often lead to reduced access to jobs, goods, services and other activities. Overtime, a reduced ability to participate in these areas of life can reduce health, wellbeing, and quality of life [6].

Research has shown that the COVD-19 public health restrictions have had the anticipated effect on reducing transmission [7,8], in part characterised by reduced mobility [9]. It has been recognised that there is a need to compare the effectiveness of different public health restrictions to provide more evidence for the current and future pandemics [10]. To facilitate surveillance of the public response to these restrictions, Google have released regular mobility reports [11]. These anonymously report on changes in human mobility at a national or a local level. They aggregate data on visits to common locations in comparison to a baseline 5-week period from 3[rd] January to 6[th] February 2020. Data is reported on six groups of locations: (I) presence at home; (II) retail and recreation; (III) grocery stores and pharmacies; (IV) public transport; (V) parks; and (VI) workplaces. This is used to generate mobility reports, whereby the percentage change from the baseline data is reported. This has been one of the main sources governments and researchers have been using to evaluate the impact of COVID-19 NPIs on behaviours.

However, no previous reviews exist that have systematically examined the impact of different public health restriction on mobility. Therefore, the aim of the current review is to assess the effect of different public health restrictions on mobility across different countries and cultures. To the authors' knowledge, this is the first study of its kind and summarising the state of our current knowledge of the effect of public health restrictions on mobility will provide information to aid decision making by policy makers and inform the direction of future researcher.

## Methods

This systematic review followed a pre-planned but unpublished protocol (available on request to corresponding author) and was conducted according to the PRISMA guidelines [12].

## Search strategy

We searched the Institute of Social and Preventive Medicine, University of Bern COVID-19 Living Evidence database (https://zika.ispm.unibe.ch/assets/data/pub/search_beta) on the 1 February 2021. This tool retrieves all COVID-19 and SARS-COV-2 publications from OVID Embase, Medline, BioRxiv and MedRxiv (details of their search available here: https://ispmbern.github.io/covid-19/living-review/collectingdata.html). The following search terms were used: ("Google" AND "Mobility") OR (global positioning) (S1 File). The titles of papers in the reference lists of included articles were screened to identify other potentially eligible studies for inclusion in the analysis missed by the initial search or any unpublished data. No additional potentially eligible papers were identified. The assessment of inclusion and exclusion criteria, quality of studies and extraction of data were independently undertaken and verified by two investigators (MAT, LMcM). The results were then compared, and, in case of discrepancies, a consensus was reached with the involvement of a third investigator (LS). There was no language restriction.

## Type of studies, inclusion and exclusion criteria

All retrospective or prospective studies evaluating the impact of COVID-19 public health restrictions on mobility were included. Mobility was measured using the Google Mobility reports, which included visits to retail outlets and parks, public transport use, number of people in workplaces and time spent in residences.

The inclusion criteria were therefore:

- Retrospective or prospective study design employed

- Assessed the effect of COVID-19 public health restrictions on mobility

- Included a measure of mobility using the Google Mobility reports

We excluded studies that did not meet the inclusion criteria.

## Data extraction

Information from included studies was extracted by one researcher (LMcM) and double checked by another (MAT). This included the characteristics of included studies, such as country, mobility data reported, dates which the study covered, and the public health restrictions in place at the time. Changes in mobility data in available variables were also extracted. Where possible, we have reported the Oxford Stringency Score [13] for the included countries, averaged across the dates covered by the study. This is a score of 0 to 100, with higher number representing stricter relative government policies. We have also reported the date of the first case in each country [14] to allow for comparison between the time frame the data covers in relation to the emergence of the pandemic.

## Risk of bias

To assess the risk of bias of included studies, the Newcastle Ottowa Scale (NOS) for non-randomised studies was used [15]. This tool assesses the risk of bias emanating from the selection of the cohort, the comparability of the findings and the measure of the primary outcome. Studies can score between zero and nine, with a higher score indicating higher quality. Studies were rated as having a high (<5), moderate (5–7) or low risk of bias (≥8) in a similar manner to previous reviews [16]. Risk of bias was assessed by one researcher (MAT) and double checked by a second (LMcM).

### Data synthesis

Given the heterogeneous nature of the designs used, a narrative synthesis was undertaken, guided by the process of Popay et al [17] and reported in accordance with the SwiM criteria for reporting systematic reviews without meta-analysis [18]. The rationale for the synthesis was to examine the effect of different COVID-19 public health restrictions of the variables from the Google Mobility reports, noting other factors that may have influenced the behavioural responses to public health restrictions. As previously stated, Google Mobility Reports contain changes in aggregated data on the visits made to six different groups of locations: (I) presence at home; (II) retail and recreation; (III) grocery stores and pharmacies; (IV) public transport; (V) parks; and (VI) workplaces. Data is reported as the percentage change from baseline. This is used to generate mobility reports, whereby the percentage change relative to the baseline period, defined as the median value, for the corresponding day of the week during the 5-week baseline period.

Preliminary synthesis of included studies was conducted by MAT and LMcM to identify common features across studies in terms of the direction and size of effect. These were tabulated and then narratively described within and across studies.

## Results

### Search results

From the initial search, 1672 references were identified (S2 File), of which 85 were selected for full text checking (Fig 1). From these 71 were excluded and 14 were included in the narrative synthesis (Fig 1).

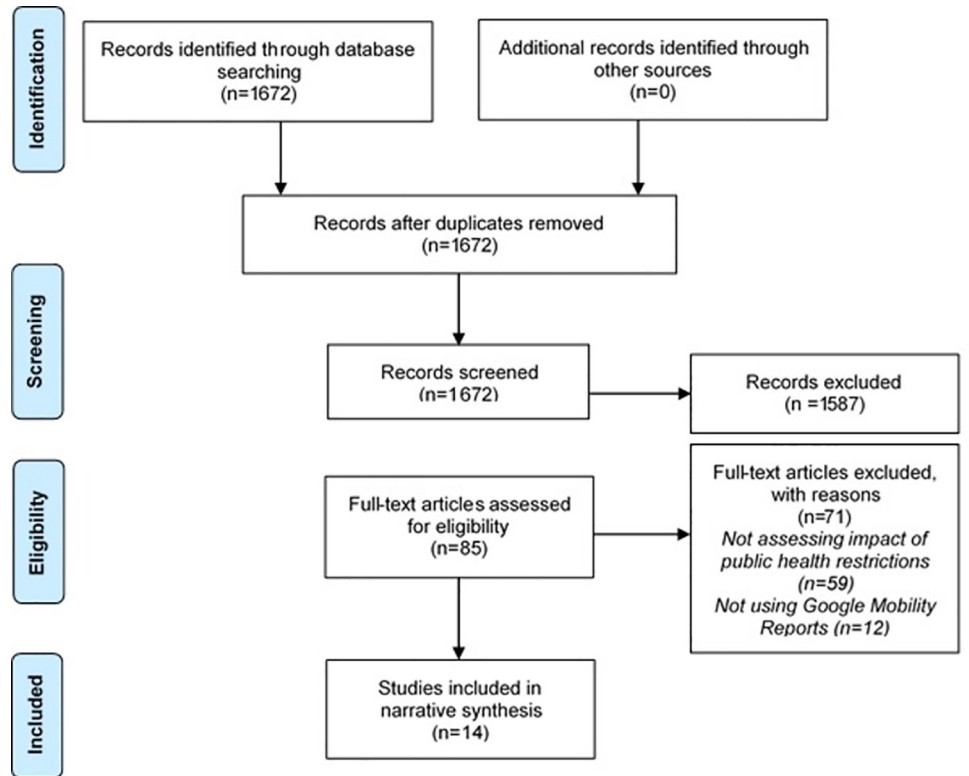

**Fig 1. PRISMA flow diagram.**

## Characteristics of included studies

Of the 14 included studies, four studies included data from multiple countries across Europe, North America, Asia, Australasia and Africa [24,28,30,32]. Four studies were conducted in the United States of America (USA) [19,21,23,25] and other studies used data from Australia [31], Canada [26], India [29], Nigeria [27], South Africa [20] and Turkey [22] (Table 1). All studies reported data from the first wave of the pandemic, with Google Mobility Scores included from January to August 2020, with most studies analysing data during the first two months of the pandemic. A range of public health restrictions were in place at the time, including stay-at-home orders, closure of non-essential services and schools and restrictions on travel within countries (Table 1). In line with this, the mean Oxford Stringency Scores ranged from 43.9 [21] to 94.2 [29] out of 100.

Changes across most or all of the Google Mobility variables where reported separately by 9 of the 14 studies [19–21,25,27,29–32], whereas four studies only reported aggregated scores score [22,23,26,28] or just assessed changes in one variable [24].

## Risk of bias

Overall, the average NOS score of included studies was 7.6 out of 9, with scores ranging from 6 to 9 (Table 2). Seven studies were rated as having a moderate risk of bias [19,20,22,23,25,31,32] and seven at a low risk of bias [21,24,26–30]. Studies that were rated as having a moderate risk of bias did not include a non-exposed comparison group or had a short follow-up duration (Table 2).

## Data synthesis

Google Mobility reports provide an estimate of change in visits to the various destinations as a percentage change from the number of visits during the baseline period of 3rd January 2020 to 6th February 2020.

## Overall Google Mobility

Wang et al [31] did not report the level of changes in the Google Mobility Scores, but identified common trends across territories in Australia including increases in the residential variable and decreased visits to transit stations, retail and recreation, and workspaces. The use of parks varied across territories and visits to grocery stores and pharmacies increased immediately after national lockdown was announced, but subsequently decreased (Table 3).

Four other studies generated an overall Google Mobility score by averaging changes in the across some or all of the variables [22,23,26,28]. In an analysis of Google Mobility score across 125 countries, stricter public health measures, measured using the Oxford COVID-19 Government Response Tracker, led to greater reductions in mobility [28], and the effects of these restrictions on mobility appeared to impact infection rates at 14 days (Table 3).

The remaining studies investigated the impact of specific restrictions. Two studies from the USA demonstrated that stay-at-home orders led to a decrease in overall mobility. Feyman et al [23] demonstrated that the introduction of stay-at-home-orders led to a decreased in overall mobility by 12 percentage points (95% confidence intervals (CI) -13.1, -10.9) over and above the average decline of -25.6 (95% CI -28.3, -22.9) that had been observed after the start of the pandemic. Excluding parks, Jacobsen & Jacobsen [25] demonstrated that the average Google Mobility score decreased by 42.5 percentage points in states with stay-at-home orders in place, compared to 32.6 percentage points in states without stay-at-home orders.

**Table 1. Characteristics of included studies.**

| Study ID | Country | Geographical Unit | Mean Oxford COVID-19 Stringency score* | Google Mobility Variables | Dates covered by the study | Date first case confirmed in country | Public Health Restrictions in place |
|---|---|---|---|---|---|---|---|
| Abouk & Heydari [19] | United States of America | 50 states plus the District of Columbia | 60.2 | (I) Presence at home (II) Retail and recreation (III) Grocery stores and pharmacies (IV) Public transport (V) Parks (VI) Workplaces | 26 March—25 April 2020 | 21 Jan 2020 | • Stay-at-home order • Closure of non-essential business and schools • Bans on large gatherings • Limits on restaurant and bars |
| Carlitz & Makhura [20] | South Africa | Provinces including Eastern Cape, Free state, Gauteng, Kwazulu-Natal, Limpopo, Mpumalanga, North West, Northern Cape, Western Cape | 87.9 | (II) Retail and recreation (III) Grocery stores and pharmacies (IV) Public transport (VI) Workplaces | 27 Mar—30 Apr 2020 | 6 Mar 2020 | • Stay-at-home order unless performing essential services, obtaining essential goods or services, collecting social grants, emergency care or chronic medication attention • Closure of non-essential businesses and workplaces except those providing essential services • Movements between provinces, metropolitan areas and districts prohibited, including commuter transport services, except for essential services |
| Chernozhukov et al [21] | United States of America | All 50 states | 43.9 | (II) Retail and recreation (III) Grocery stores and pharmacies (IV) Public transport (VI) Workplaces | 7 March—3 June 2020 | 21 Jan 2020 | • Stay-at-home order • Closure of K-12 schools, movie theatres, dine in restaurants and non-essential businesses • Mandatory face mask use by employees |
| Durmuş et al [22] | Turkey | - | 70.5 | Aggregated Google Mobility Score | 11 March—18 April 2020 | 12 Mar 2020 | • Ban on flights and restrictions on inter-provincial travel • Closure of schools • Postponements of sporting events • Introduction of a curfew for those under 20 years and over 65 years of age |
| Feyman et al [23] | United States of America | All 50 states | 52.1 | Mobility index calculated as mean of percent changes for all non-residential categories, which included retail and recreation, groceries and pharmacies, parks, transit stations, and workplaces | 19 March—7 April 2020 | 21 Jan 2020 | • Stay at home order implemented in 39 states except for certain permitted activities, e.g. key workers and shopping for essential supplies |

(*Continued*)

**Table 1.** (*Continued*)

| Study ID | Country | Geographical Unit | Mean Oxford COVID-19 Stringency score* | Google Mobility Variables | Dates covered by the study | Date first case confirmed in country | Public Health Restrictions in place |
|---|---|---|---|---|---|---|---|
| Geng et al [24] | Argentina, Australia, Austria, Belgium, Bolivia, Brazil, Canada, Chile, Colombia, Denmark, Ecuador, Egypt, Finland, France, Germany, Hong Kong, Hungary, India, Indonesia, Ireland, Italy, Japan, Kenya, Malaysia, Mexico, Mongolia, Netherlands, New Zealand, Nigeria, Norway, Panama, Peru, Philippines, Poland, Portugal, Romania, Saudi Arabia, Singapore, South Africa, South Korea, Spain, Sweden, Taiwan, Thailand, UK, United States, Vietnam. | 48 countries across all continents | - | (V) Parks | 16 February—26 May 2020 | - | • Stay-at-home order<br>• Restrictions on social gatherings, and internal movements<br>• Cancellation of public events<br>• Closure of workplaces |
| Jacobsen & Jacobsen [25] | United States of America | 50 states plus the District of Columbia | 60.2 | (II) Retail and recreation<br>(III) Grocery stores and pharmacies<br>(IV) Public transport<br>(V) Parks<br>(VI) Workplaces | 29 March 2020 | 21 Jan 2020 | • Stay-at-home order except for essential activities such as key work, exercise and shopping for food introduced in 25 states<br>• Work from home order, except for key workers who are unable to do so |
| Karaivanov et al [26] | Canada | - | 61.5 | (I) Presence at home<br>(II) Retail and recreation<br>(III) Grocery stores and pharmacies<br>(IV) Public transport<br>(V) Parks<br>(VI) Workplaces | 26 February 2020–3 July 2020 | 26 Jan 2020 | • Mandatory mask wearing<br>• Closure of retail and non-essential businesses, restaurants, recreation facilities, places of worship and schools.<br>• Limits on events and gatherings<br>• Restrictions on international and domestic travel |
| Lawal & Nwegbu [27] | Nigeria | | 83.4 | (I) Presence at home<br>(II) Retail and recreation<br>(III) Grocery stores and pharmacies<br>(IV) Public transport<br>(V) Parks | 29 March—30 June 2020 | 28 Feb 2020 | • Nationwide total lockdown |

(*Continued*)

**Table 1.** (Continued)

| Study ID | Country | Geographical Unit | Mean Oxford COVID-19 Stringency score* | Google Mobility Variables | Dates covered by the study | Date first case confirmed in country | Public Health Restrictions in place |
|---|---|---|---|---|---|---|---|
| Ould Setti & Vountilainen [28] | 125 countries | | - | (I) Presence at home (II) Retail and recreation (III) Grocery stores and pharmacies (IV) Public transport (V) Parks (VI) Workplaces | 15 February —11 September 2020 | - | Stay-at-home order • Social distancing • Working from home • Shopping only for essentials and leaving residency only for essential reasons, such as key work or seeking medical care. |
| Singh et al [29] | India | | 94.2 | (I) Presence at home (II) Retail and recreation (III) Grocery stores and pharmacies (IV) Public transport (V) Parks (VI) Workplaces | 22 Mar—17 May 2020 | 30 Jan 2020 | • Curfew • Restricted inter-state movement • Closure of non-essential services and schools |
| Summan & Nandi [30] | 130 countries across: East Asia & Pacific; Europe & Central Asia; Latin America & Caribbean; Middle East & North Africa; North America; South Asia; and Sub-Saharan Africa | | - | (I) Presence at home (II) Retail and recreation (III) Grocery stores and pharmacies (IV) Public transport (V) Parks (VI) Workplaces | Mar—April 2020 | - | **Standard lockdown** • Stay-a-home order except for essential activities • Closure of all non-essential business and **Strict lockdown** • Closure of all industries, except for those deemed essential • Individuals only allowed to leave home for essential activities • Curfew which allowed people to leave home at specific times • Fines issued if individuals not complying • Military presence to enforce measures |
| Wang et al [31] | Australia | South Australia; West Australia; Tasmania; North Territory; Australian Capital Territory; New South Wales; Victoria; Queensland | 58.2 | (I) Presence at home (II) Retail and recreation (III) Grocery stores and pharmacies (IV) Public transport (V) Parks (VI) Workplaces | 15 Feb—15 Aug 2020 | 25 Jan 2020 | • Travel restrictions • Self-isolation • Social distancing • Border closures |
| Xu [32] | United States of America and Europe | | - | (I) Presence at home (II) Retail and recreation (III) Grocery stores and pharmacies (IV) Public transport (V) Parks (VI) Workplaces | Not reported (assumed to be during first wave) | - | • Stay-at-home order in the USA • Lockdowns in Europe with similar public health restrictions such as stay-at-home orders with the exception of essential activities |

*The stringency score across multiple countries have not been included.

**Table 2. Risk of bias of included studies.**

| Study ID | Selection | | | | Comparability | | Outcome | | | Total score (/9) |
|---|---|---|---|---|---|---|---|---|---|---|
| | Representativeness of the exposed cohort | Selection of the non-exposed cohort | Ascertainment of exposure | Demonstration that outcome of interest was not present at start of study | Comparability of cohorts on the basis of the design or analysis | Comparability of cohorts on the basis of the design or analysis | Assessment of outcome | Was follow-up long enough for outcomes to occur | Adequacy of follow up of cohorts | |
| Abouk & Heydari [19] | 1 | 0 | 1 | 1 | 1 | 1 | 1 | 0 | 1 | 7 |
| Carlitz & Makhura [20] | 1 | 1 | 1 | 1 | 1 | 0 | 1 | 0 | 1 | 7 |
| Chernozhukov et al [21] | 1 | 1 | 1 | 1 | 1 | 1 | 1 | 1 | 1 | 9 |
| Durmuş et al [22] | 1 | 0 | 1 | 1 | 0 | 0 | 1 | 1 | 1 | 6 |
| Feyman et al [23] | 1 | 1 | 1 | 1 | 1 | 0 | 1 | 0 | 1 | 7 |
| Geng et al [24] | 1 | 1 | 1 | 1 | 1 | 0 | 1 | 1 | 1 | 8 |
| Jacobsen & Jacobsen [25] | 1 | 1 | 1 | 1 | 1 | 0 | 1 | 0 | 1 | 7 |
| Karaivanov et al [26] | 1 | 1 | 1 | 1 | 1 | 1 | 1 | 1 | 1 | 9 |
| Lawal & Nwegbu [27] | 1 | 1 | 1 | 1 | 1 | 0 | 1 | 1 | 1 | 8 |
| Ould Setti & Vountilainen [28] | 1 | 1 | 1 | 1 | 1 | 0 | 1 | 1 | 1 | 8 |
| Singh et al [29] | 1 | 1 | 1 | 1 | 1 | 0 | 1 | 1 | 1 | 8 |
| Summan & Nandi [30] | 1 | 1 | 1 | 1 | 1 | 1 | 1 | 1 | 1 | 9 |
| Wang et al [31] | 1 | 0 | 1 | 1 | 0 | 0 | 1 | 1 | 1 | 6 |
| Xu [32] | 1 | 1 | 1 | 1 | 1 | 0 | 1 | 0 | 1 | 7 |

In Turkey, Durmuş et al [22] demonstrated that the introductions of stay-at-home orders for people aged over 65 years, travel restrictions, closure of schools & cancellation of major social activities led to a decrease of 36 percentage points. They also found that the most effective strategy (84.8 percentage point decrease) was the introduction of stay-at-home orders for all people in the country for two days, leading to the reduction in the virus transmission rate from 7.52 to 1.82. and a significant positive correlation was found between Google Mobility data and transmission rate (r = 0.78).

In Canada, restrictions on international and domestic travel, and on visiting care facilities, were weakly correlated (r = 0.14) with a reduction in the Google Mobility score, whereas the restrictions on non-essential businesses was strongly correlated (r = -0.86) with a reduction in the Google Mobility score. Although the introduction of mask mandates appeared to have an impact on transmission rates, it did not appear to be correlated with mobility (r = 0.09) [26].

## Presence at home

One of the more common public health measures introduced during the first wave of the pandemic was the introduction of stay-at-home orders. This was shown to result in an increase in the presence at home Google Mobility score, ranging from 16.2 percentage points in the USA [19] to 20.6 percentage points in Europe [32] and 29 (95% CI 17–32) percentage points increase in India [29]. The variation in regional effects was noted in a study in Nigeria [27]

**Table 3. Effects of public health restrictions of Google Mobility variables.**

| Study ID | Change in Overall Mobility Score (percentage points) | (I) Presence at home | (II) Retail and Recreation | (III) Grocery stores and pharmacies | (IV) Public Transport | (V) Parks | (VI) Workplace |
|---|---|---|---|---|---|---|---|
| Abouk & Heydari [19] | - | +16.2%<br>• State-wide stay-at-home orders: +15.2%<br>• Limits on restaurants and bars: + 8.5% | -36.9%<br>• State-wide stay-at-home order: -13%<br>• Limits on restaurants and bars -11% | -6.2%<br>• State-wide stay-at-home order: -110%<br>• School closure: -22%<br>• Limits on restaurants and bars: -32% | -40.9%<br>• State-wide stay-at-home order: -19%<br>• Limits on restaurants and bars: -17% | • Mobility in parks 7.3%<br>• State-wide stay-at-home order -143%<br>• Limits on restaurants and bars -163% | • Workplace -40.5%<br>• State-wide stay-at-home order -13%<br>• Limits on restaurants and bars -7% |
| Carlitz & Makhura [20] | - | - | -71% | -46% | -71% | - | - 60% |
| Chernozhukov et al [21] | - | - | Correlation between policies and weekly changes in mobility:<br>• mask mandate = -0.17<br>• stay-at-home orders = -0.69<br>• closure of schools = −0.79<br>• -closure of non-essential businesses, movie theatres and restaurants = −0.84 | Correlation between policies and weekly changes in mobility:<br>• mask mandate = -0.15<br>• stay-at-home orders = −0.70<br>• closure of schools = −0.55<br>• closure of non-essential businesses, movie theatres and restaurants = −0.75 | Correlation between policies and weekly changes in mobility:<br>• mask mandate = -0.29<br>• stay-at-home orders = −0.71<br>• closure of schools = −0.72<br>• closure of non-essential businesses, movie theatres and restaurants = −0.79 | - | Correlation between policies and weekly changes in mobility:<br>• mask mandate = -0.32<br>• stay-at-home orders = −0.69<br>• closure of schools = −0.91<br>• closure of non-essential businesses, movie theatres and restaurants = −0.84 |
| Durmuş et al [22] | Mean (SD): -36.33 (22.41) | - | - | - | - | - | - |
| Feyman et al [23] | *Before shelter-in place order Mean (95% CI) -25.6 (-28.3, -22.9). Additional effect after shelter-in place order -12 (-13.1, -10.9)* | - | - | - | - | - | - |
| Geng et al [24] | - | - | - | - | - | In stepwise regression, stay at home restrictions were independently associated with reductions in park use (std. coefficient β = − 0.341, p < 0.001). By contrast, social gathering restrictions (std. coefficient β = 0.19, p < 0.001), public event cancellations (std. coefficient β = 0.126, p < 0.001), workplace closures (std. coefficient β = 0.092, p = 0.001) and movement restrictions (std. coefficient β = 0.048, p = 0.039) were independently associated with increased park use. | - |
| Jacobsen & Jacobsen [25] | • States without a stay-at-home orders: -32.6<br>• States with stay-at-home orders: -42.5% | - | • States without stay-at-home orders: −41.2%<br>• States with stay-at-home orders: −51.3% | • States without stay-at-home orders: −15.5%<br>• States with stay-at-home orders: −26.9% | • States without stay-at-home orders: −39.6%<br>• States with stay-at-home orders: −53.2% | • States without stay-at-home orders: +25.8%<br>• States with stay-at-home orders: −9.7% | • States without stay-at-home orders: −33.9%<br>• States with stay-at-home orders: −38.4% |
| Karaivanov et al [26] | Correlation between policies and weekly changes in mobility:<br>-mask mandate = 0.09<br>-restrictions on non-essential businesses = -0.86<br>-school closure = -0.37<br>-international and domestic travel restrictions = -0.14<br>-restrictions on visiting long-term care facilities = -0.14 | - | - | - | - | - | - |
| Lawal & Nwegbu [27] | | Most (65%) states recorded a 10% increase or more | Most States recorded a median mobility decline ranging between 10 and 44% and no statistically significant trend over time. | Most States recorded a median mobility decline ranging between 10 and 39% and a statistically significant upward trend over time | Most States recorded a median mobility decline ranging between 3 and 75% with variation in the trend over time across states | Most States recorded a median mobility decline ranging between 10 and 39% and no statistically significant trend over time. | Most States recorded a decline ≥10% and a statistically significant upward trend over time |
| Ould Setti & Vountilainen [28] | Correlation between COVID stringency index and mobility r = 0.7 | - | - | - | - | - | - |

(*Continued*)

**Table 3.** (Continued)

| Study ID | Change in Overall Mobility Score (percentage points) | (I) Presence at home | (II) Retail and Recreation | (III) Grocery stores and pharmacies | (IV) Public Transport | (V) Parks | (VI) Workplace |
|---|---|---|---|---|---|---|---|
| Singh et al [29] | - | Median: 29% (95% CI 17–32) | Overall median 69% decrease (95% CI 54–87) | Overall median 47% decrease (95% CI 22–76) | Overall median 64% decrease (95% CI 52–74) | Overall median 58% decrease (95% CI 35–68) | Overall median 62% decrease (95% CI 27–72) |
| Summan & Nandi [30] | - | **Short follow-up period (2 days after implementation)**<br>• Standard Lockdown 30 days after first case = 20.61% (95%CI 7.57, 33.65)<br>• Standard Lockdown 45 days after first case = 39.48% (95%CI 26.91, 52.04)<br>• Strict lockdown 30 days after first case = 30.12% (95%CI 15.20, 45.03)<br>• Strict lockdown 45 days after first case = 35.03% (95%CI 20.49, 49.58)<br>**Longer follow-up period (6 days after implementation)**<br>• Standard Lockdown 30 days after first case = 141.44% (95% CI 71.79, 210.48)<br>• Standard Lockdown 45 days after first case = 198.65% (95% CI 133.05, 264.24)<br>• Strict lockdown 30 days after first case = 203.15% (95% CI 127.41, 278.88)<br>• Strict lockdown 45 days after first case = 219.28% (95% CI 149.96, 288.60) | **Short follow-up period (2 days after implementation)**<br>• Standard Lockdown 30 days after first case = -14.81% (95%CI -23.54, -6.07)<br>• Standard Lockdown 45 days after first case = -31.36% (95%CI -40.99, -21.73)<br>• Strict lockdown 30 days after first case = -19.73% (95%CI -30.99, -8.48)<br>• Strict lockdown 45 days after first case = -26.65% (95%CI -38.98, -14.31)<br>**Longer follow-up period (6 days after implementation)**<br>• Standard Lockdown 30 days after first case = -7.77% (95%CI -12.14, -3.14)<br>• Standard Lockdown 45 days after first case = -15.84% (95%CI -22.26, -9.42)<br>• Strict lockdown 30 days after first case = -10.14% (95%CI -16.20, -4.07)<br>• Strict lockdown 45 days after first case = -15.53% (95%CI -24.23, -6.82) | **Short follow-up period (2 days after implementation)**<br>• Standard Lockdown 30 days after first case = -8.80% (95%CI -23.44, 5.84)<br>• Standard Lockdown 45 days after first case = -29.65% (95%CI -44.4, -14.9)<br>• Strict lockdown 30 days after first case = -18.83% (95%CI -35.31, -2.36)<br>• Strict lockdown 45 days after first case = -25.84% (95%CI -43.36, -8.32)<br>**Longer follow-up period (6 days after implementation)**<br>• Standard Lockdown 30 days after first case = -16.90% (95%CI -27.80, -6.01)<br>• Standard Lockdown 45 days after first case = -39.46% (95%CI -51.17, -27.75)<br>• Strict lockdown 30 days after first case = -30.46% (95%CI -47.74, -13.18)<br>• Strict lockdown 45 days after first case = -41.69% (95%CI -62.44, -20.94) | **Short follow-up period (2 days after implementation)**<br>• Standard Lockdown 30 days after first case = -14.21% (95%CI -22.59, -5.84)<br>• Standard Lockdown 45 days after first case = -32.03% (95%CI -41.60, -22.45)<br>• Strict lockdown 30 days after first case = -19.11% (95%CI -30.11, -8.11)<br>• Strict lockdown 45 days after first case = -26.97% (95%CI -39.05, -14.88)<br>**Longer follow-up period (6 days after implementation)**<br>• Standard Lockdown 30 days after first case = -7.23% (95%CI -11.79, -2.67)<br>• Standard Lockdown 45 days after first case = -16.17% (95%CI -23.22, -9.11)<br>• Strict lockdown 30 days after first case = -9.62% (95%CI -16.32, -2.91)<br>• Strict lockdown 45 days after first case = -14.26% (95%CI -23.17, -5.34) | **Short follow-up period (2 days after implementation)**<br>• Standard Lockdown 30 days after first case = -11.31% (95%CI -35.51, 12.89)<br>• Standard Lockdown 45 days after first case = -21.33% (95%CI -42.67, 0.01)<br>• Strict lockdown 30 days after first case = -36.08% (95%CI -65.01, -7.15)<br>• Strict lockdown 45 days after first case = -36.34% (95%CI -63.19, -9.49)<br>**Longer follow-up period (6 days after implementation)**<br>• Standard Lockdown 30 days after first case = 4.42% (95%CI -104.60, 113.44)<br>• Standard Lockdown 45 days after first case = 12.46% (95%CI -88.57, 113.49)<br>• Strict lockdown 30 days after first case = -124.14% (95%CI -293.13, 44.85)<br>• Strict lockdown 45 days after first case = -95.15% (95%CI -252.07, 61.78) | **Short follow-up period (2 days after implementation)**<br>• Standard Lockdown 30 days after first case = -18.21% (95%CI -27.69, -8.73)<br>• Standard Lockdown 45 days after first case = -38.60% (95%CI -49.40, -27.80)<br>• Strict lockdown 30 days after first case = -25.07% (95%CI -37.94, -12.20)<br>• Strict lockdown 45 days after first case = -31.85% (95%CI -45.95, -17.75)<br>**Longer follow-up period (6 days after implementation)**<br>• Standard Lockdown 30 days after first case = -14.23% (95%CI -21.24, -7.23)<br>• Standard Lockdown 45 days after first case = -26.62% (95%CI, (-35.70, -17.54)<br>• Strict lockdown 30 days after first case = -20.44% (95%CI -32.14, -8.74)<br>• Strict lockdown 45 days after first case = -26.49% (95%CI -40.68, -12.31) |
| Wang et al [31] | Overall changes in mobility indices not reported. Mobility decreased within three days of the introduction of restrictions. After the restrictions were eased, mobility increased. | Increased after the introduction of restrictions | Decreased after the introduction of restrictions | Initially increased after the introduction of restrictions due to panic-buying | Decreased after the introduction of restrictions | Decreased after the introduction of restrictions, but started to increase again after 1 month | Decreased after the introduction of restrictions, but started to increase again after 1 month |
| Xu [32] | - | USA: 17.5% Europe: 20.59% | USA: -44.66% Europe: -64.38% | USA: -14.77% Europe: -26.84% | USA: -45.33% Europe: -62.66% | USA: -2.57% Europe: -11.8% | USA: -45.05% Europe: -51.66% |

Abbreviations: SD-standard deviation, C = confidence intervals.

where there was wide variation across states, with most (65%) recording at least a 10 percentage point increase.

The impact of specific stay-at-home orders in the USA led to a significant increase in the presence at home score of 15.2 percentage points and limitations on restaurants and bars increased presence at home by 8.5 percentage points [19].

In a study of 130 countries, standard restrictions (closure of all non-essential business and stay-a-home orders except for essential activities) led to a short-term increase of 20.6 (95%CI 7.6, 33.7) and 39.5 (95%CI 26.9, 52.0) percentage points after 30 and 45 days respectively [30]. This was somewhat lower than the effect in countries that had introduced stricter restrictions (all industries, except for those deemed essential, completely closed, individuals only allowed to leave home for essential activities, a curfew which allowed people to leave home at specific

times, fines issued if individuals not complying and military presence to enforce measures). In these countries, the presence at home was increased by 35.1 (95%CI 15.20, 45.03) percentage points 30 days after the first case, but did not reach the level of the standard lockdown measures after 45 days (35 percentage point increase; 95%CI 20.49, 49.58). However, the differences between standard and strict lockdowns at 45 days were similar to the differences at 30 days when data was assessed 6 days after the implementation of restrictions.

### Retail and recreation

Visits to retail and recreation destinations were considerably reduced (Table 3), in keeping with the closure of non-essential businesses in many countries. This ranged between 37 percentage points reduction in USA [19] to 71 percentage points reduction in South Africa. In the USA, states with stay-at-home orders demonstrated a further reduction in visits to retail and recreations destinations than those that did not [25]. Chernozhukov et al [21] identified that the introduction of stay-at-home orders, the closure of schools, non-essential businesses, movie theatres and restaurants were moderately to strongly correlated with decreases in visits to retail and recreation destinations.

In a similar way to the presence at home variable, countries with stricter restrictions experienced a greater decline in visits to retail and recreation in the first 30 days when compared to those with standard restrictions (decrease of 19.7 percentage points vs 14.8 percentage points) [30]. However, this was reversed after 45 days (26.7 percentage point decrease in countries with strict restrictions vs 31.4 percentage points in countries with standard restrictions) (Table 3). The differences between standard and strict lockdowns at 45 days were negligible when data was assessed 6 days after the implementation of restrictions (Table 3).

### Grocery stores and pharmacies

Visits to grocery stores and pharmacies were also reduced (Table 3), albeit not consistently to the same level, with reductions ranging from 6 percentage points in the USA [19] to 71 percentage points in South Africa [20]. The introduction of stay-at-home orders in certain states in the USA had a greater effect than those without in terms of the reduction of visits to grocery stores and pharmacies. (decrease of 27 percentage points vs 16 percentage points) [25]. In the USA, identified that the introduction of stay-at-home orders, the closure of schools, non-essential businesses, movie theatres and restaurants were moderately to strongly correlated with reduced visits to grocery stores and pharmacies [21]. Xu [32] identified greater declines in visits to grocery stores and pharmacies in Europe (26.8 percentage points) compared to the USA (14.8 percentage points). Finally, countries with stricter restrictions experienced a greater decline in visits to grocery stores & pharmacies 30 days and 45 days (longer term data) after the first case, as shown in an analysis of data from multiple countries [30].

### Public transport

Alongside the closure of workplaces and the introduction of stay-at-home orders, the use of public transport declined by between 41 [19] and 71 percentage points [20] (Table 3). There were only small differences in the effect of different public health restrictions on public transport use. For example, Abouk & Heydari [19] identified that stay-at-home orders led to an additional decline in public transport of 19 percentage points, compared with 17 percentage points from the closure of restaurants and bars. Chernozhukov et al demonstrated that the introduction of stay-at-home orders, the closure of schools, non-essential businesses, movie theatres and restaurants were moderately correlated with decreases public transport use, whereas the introduction of mask mandates were weakly correlated with public transport use

[21]. In contrast to this, Jacobsen & Jacobsen [25] demonstrated that public transport use decreased by 53 percentage points in states with stay-at-home orders but by 40 percentage points in states without. Similarly, countries that implemented strict restrictions experienced greater declines in public transport use compared to countries with standard restrictions (Table 3) [30].

### Parks

One study specifically examined the impact of public health restrictions on park use. In a study of data from 48 countries, Geng et al [24] demonstrated that the introduction of stay-at-home orders was independently associated with reduction in park use. By contrast, restrictions on social gatherings and travel, the cancellation of public events and the closure of workplaces were independently associated with increased park use. Similar findings of the effects of stay-at-home orders on park use were demonstrated by other studies [19,25]. Summan & Nandi [30] demonstrated that countries with stricter restrictions experienced a greater decline in visits to parks after both 30 and 45 days, which was not the case for other mobility variables (Table 3). Of note, two studies did report on changes in park use due to uncertainty as to whether visiting a park presented an increased risk of transmission or not [21,28].

### Workplace

As with the other variables, there was some variation in the impact of the COVID-19 public heath restrictions on visits to the workplace (Table 3), with effects ranging from a 41 percentage point reduction in the USA [19] to a 62 percentage point reduction in India [29], reflecting a trend in the USA to gradually return to the workplace over time [27]. Countries with stricter public health restrictions experienced greater reductions in visits to workplaces [30]. The introduction of stay-at-home orders, the closure of schools, non-essential businesses, movie theatres and restaurants were moderate to strongly correlated with decreases in visits to workplaces [21].

## Discussion

This systematic review demonstrates that mobility was significantly impacted by the public health restrictions put in place to reduce the transmission during the first wave of the COVID-19 pandemic. There was moderate to good quality evidence of an effect across each of the Google Mobility variables, though there was considerable variety across countries. Areas that introduced stay-at-home orders had a consistently larger effect on mobility. By contrast, mask mandates had little to no apparent effect on mobility.

Of note, the study by Summan & Nandi [30] identified a differential impact on mobility in countries that introduced stricter public health measures. As expected, stricter lockdowns led to greater reductions across the mobility variables in the first 30 days after the detection of the first case and six days after the implementation of restrictions. However, the large-scale, high-quality study by Summan & Nandi [30], which utilised data from 130 countries across all continents, also demonstrated that this trend was not the same when shorter-term data was assessed. Changes in Google Mobility was higher in countries with standard lockdowns compared to more stringent ones when the data was analysed within the first two days after restrictions were implemented. There a few reasons why this might be the case. There could be a ceiling effect of standard restrictions that mean to achieve greater levels of behavioural responses requires more stringent methods. The authors also noted that national lockdowns may not be feasible in poorer countries where broader support systems do not exist and a

greater proportion of the population receive daily wages, so are required to work. The differences observed may therefore relate to the socio-economic status of the country [33].

There was some uncertainty across included studies around of the risk of transmission from visiting parks, and subsequent mixed findings on the impact of public health restrictions on park use. Evidence has shown that the exposure to green space, such as parks, is beneficial to mental health and wellbeing [34]. Given that the pandemic has led to increased symptoms of stress and anxiety [35], the therapeutic benefits of the use of green space should not be underestimated and many people may have sought this whilst spending more time at home. Transmission of COVID-19 has been reported in parks, but a very low (<10%) proportion of infections occur outdoors, and those that do tend to involve large gatherings and occasional indoor gatherings [36]. In general, simple mitigations such as limiting the duration and frequency of personal contact and wearing a face covering if in close physical proximity can effectively mitigate this risk [37].

In contrast to the effectiveness of restrictions that directly impacted movement, such as stay at home orders, the introduction of mask mandates did not appear to influence mobility. NPIs such as masks [21,26] may be required beyond the acute phases of the pandemic and to prevent future respiratory pandemics [38]. Whilst we acknowledge the potential side effects of ongoing use of face coverings, such has inhibited communication [39], adopting the use of face coverings as a social practice, in a similar manner to some Asian countries in response to previous respiratory pandemics, would seem beneficial [40]. Evidence from the current review is reassuring in that while mask mandates reduced transmission, they did not significantly impact mobility. This suggest that recommending their ongoing use should not inhibit the return to previous patterns of daily life.

In many countries, the use of public transport was not prevented, but discouraged during the first wave of the pandemic, leading to sustained decreases in public transport use. While this restriction was required to prevent transmission, habits are likely to have been broken leading to prolonged reluctance to use public transport as restrictions are lifted. Given the individual, societal and environmental benefits of public transport use, there is a need to plan for how to encourage people back on to public transport, supporting social distancing and rebuilding confidence [41].

A limitation of the included studies was that they reported data exclusively from the first wave of the pandemic. Indeed, many of the studies reported data from the early phase of the first wave. This was likely fuelled with a desire to produce early research findings to inform practice and policy, but may not represent the full extent of restrictions in place, so the findings should be interpreted accordingly. Further research is required to determine the impact of the reintroduction of public health restrictions in the subsequent autumn/winter waves that some countries experienced. Warnings have been given that populations would not be able to sustain transmission prevention behaviours, such as reduced mobility [42]. This has become known as 'pandemic fatigue', and has been cited as one of the reasons for delaying the introduction of more strict restrictions in the UK in response to the second wave [43]. A cross-sectional study of adults in Australia, USA and UK showed that although adherence to restrictions remained high into the second wave, it did decline, particularly in young people and males [44]. In addition, there are limitations with the Google Mobility data itself. The data is a proxy for 'real' observations and only represents users of Android phones who have given permission for Google to utilise their location data. Therefore, it may not be fully representative of the whole population, or provide data on countries where google coverage is limited or where mobile data coverage isn't consistent.

Another limitation in the review is the inclusion of preprint papers. A notable feature of science communication during the COVID-19 pandemic was the use of preprint servers to

facilitate the rapid dissemination of relevant research [45]. These papers have not been through a rigorous peer review process and therefore their conclusions may be altered in response to feedback. As such, the findings of this review should be interpreted accordingly. However, the included studies were all rated as fair to moderate quality and the findings were relatively consistent across studies, so the overall conclusions of the review should be unaffected.

## Conclusion

In response to the COVID-19 pandemic, public health restrictions, particularly stay-at-home orders have significantly impacted on transmission prevention behaviour. Although these NPIs have successfully altered behaviour, further research is required to understand how to effectively address pandemic fatigue, especially in young people and males, and to support the safe return back to normal day-to-day behaviour. Targeted transport policies and safety measures can also enhance the mobility of younger people and encourage use of public transport as countries recover from COVID-19. It is vital that considerations of health equity and social justice principles remain at the forefront of pandemic response strategies and NPIs. The review has also highlighted the value of partnering with multinational technology companies to monitor and tract the effects of public health policy in real time and for cross-country comparisons.

## Supporting information

**S1 Checklist. PRISMA 2009 checklist.**
(DOCX)

**S1 File. COVID-19 living evidence database search.**
(DOCX)

**S2 File. Search results.**
(XLSX)

## Author Contributions

**Conceptualization:** Mark A. Tully, Deepti Adlakha, Neale Blair, Jonny McAneney, Helen McAneney, Christina Carmichael, Conor Cunningham, Nicola C. Armstrong, Lee Smith.

**Data curation:** Mark A. Tully, Laura McMaw.

**Formal analysis:** Laura McMaw.

**Methodology:** Mark A. Tully.

**Project administration:** Mark A. Tully.

**Supervision:** Mark A. Tully.

**Writing – original draft:** Mark A. Tully.

**Writing – review & editing:** Mark A. Tully, Laura McMaw, Deepti Adlakha, Neale Blair, Jonny McAneney, Helen McAneney, Christina Carmichael, Conor Cunningham, Nicola C. Armstrong, Lee Smith.

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
