## [Decision Letter · Decision Letter 0]

23 Aug 2021

PONE-D-21-23011

The effect of different COVID-19 public health restrictions on mobility: a systematic review

PLOS ONE

Dear Dr. Tully,

Thank you for submitting your manuscript to PLOS ONE. After careful consideration, we feel that it has merit but does not fully meet PLOS ONE’s publication criteria as it currently stands. Therefore, we invite you to submit a revised version of the manuscript that addresses the points raised during the review process.

We look forward to receiving your revised manuscript.

Kind regards,

Sanjay Kumar Singh Patel, Ph.D.

Academic Editor

PLOS ONE

Journal Requirements:

Reviewers' comments:

Reviewer's Responses to Questions

**Comments to the Author**

1. Is the manuscript technically sound, and do the data support the conclusions?

Reviewer #1: Partly

Reviewer #2: Yes

2. Has the statistical analysis been performed appropriately and rigorously? 

Reviewer #1: Yes

Reviewer #2: Yes

3. Have the authors made all data underlying the findings in their manuscript fully available?

Reviewer #1: Yes

Reviewer #2: Yes

4. Is the manuscript presented in an intelligible fashion and written in standard English?

Reviewer #1: Yes

Reviewer #2: Yes

5. Review Comments to the Author

Reviewer #1: I have uploaded a separate file named "comments". please refer to that for detailed explanations. This is a nicely written review which can be improved on. i haven't found any major flaws in the language barring few minor typos.

Reviewer #2: The manuscript by Tully et al. “The effect of different COVID-19 public health restrictions on mobility: a systematic review” is well-organized and presented. This manuscript can be accepted in PloS One after minor revision as follows:

Comments

1. Lines 60-70, please elaborate what kinds of various measures? In addition to testing and quarantining the uses of some traditional medicine -based biomolecules as anti-covid 19 agents were founded effective and few drugs also in absence of vaccine i.e. doi: 10.1007/s12088-020-00893-4. In addition, diet and gut microbiota also played important role in combating such infections i.e. doi: 10.1007/s12088-020-00908-0. Please add minor summary of such information’s.

2. Please highlight the clear objective and significance of this article at the end of Introduction.

3. Tables may be reorganized more specifically in the term of text contents.

---

## [Author Response · Author response to Decision Letter 0]

13 Oct 2021

Overall

Thank you for the taking the time to read and review our manuscript. We are very grateful for the positive comments and suggestions for minor amendments. We have responded to each comment below and believe the manuscript has improved as a result. We trust these changes are acceptable.

Reviewer 1 

Although this is a good review however as the authors rightly pointed out that there are limitations with the Google Mobility data itself. I strongly believe that this Google Mobility data doesn’t represent the accurate picture. 

Response: Thank you for your helpful comments on our review. We appreciate your opinion on the accuracy of the Google Mobility data. We had previously noted the limitations of the data, though it should be noted that a number of countries are using this data in their pandemic modelling and so we still believe that there is value in reviewing papers that have utilized it. 

Line 41 to 44: the author says most studies analyzing data during the first two month i.e January and February, of pandemic. 

And during this time not all country introduced stringent restrictions to human movements. So how is this study not limited to low set of data. 

Response: The search for the review was conducted in February 2021, therefore we did not purposefully restrict our data to that time period. It is just the time period from which the included studies covered. We have noted that the included studies only cover the early period so that the reader can understand the limitations and interpret the evidence accordingly. To further contextualize the results, we have noted the stringency index that corresponds with the time the data was collected, and have included a note on what restrictions were in place at the time the data was collected.

We have added to the limitations to acknowledge that the data available at the time covered the early phase of the first wave of the pandemic and the evidence should be interpreted accordingly.

Line 84 to 90: To facilitate surveillance of the public response to these restrictions, Google have released regular mobility reports. These anonymously report on changes in human mobility at a national or a local level.

I am wondering how exhaustive these data are. This is because as compared to big nations like US, UN or Canada large chunks of small and underdeveloped countries don’t have a good access of internet. Also, some countries don’t use google at all. Then does this means google is not considering them pocket of populations? It would be nice if the author can comment on this. 

Response: That is an interesting perspective and we appreciate this insight. We have added a comment on this to the limitations.

Line 108 to 109: The reference lists of included articles were hand-searched to identify other potentially eligible studies for inclusion in the analysis missed by the initial search or any unpublished data. 

Can you elaborate what do you mean by hand-search. What are the criteria for identifying the potential study using this method?

Response: We have clarified the process. The titles in reference lists were screened for potentially eligible papers. None were identified through this process.

Line 159 to 160: From the initial search, 1672 references were identified, of which 85 were selected for full text checking (Figure 1). From these 71 were excluded and 14 were included in the narrative synthesis (Figure 1).

It’s a very nice flowchart, however it’s not clear to me the rationale of screening and eligibility. It would be nice if the author includes few lines describing the criteria, they have used to screen studies for their narrative synthesis.

Response: As is standard practice for systematic reviews, the eligibility criteria used when screening articles were defined a priori. They are described in the methods, within the section “Type of studies, inclusion and exclusion criteria”. 

We have added text to this section to make it more explicit what the inclusion criteria were.

Line 380 to 382: Given the individual, societal and environmental benefits of public transport use, there is a need to plan for how encourage people back on to public transport, supporting social distancing and rebuilding confidence

Correct the typo between line 380-381. 

Response: Thank you for bringing this to our attention. We have now amended the typo.

Line 380 to 382: A limitation of the included studies was that they reported data exclusively from the first wave of the pandemic.

Did first wave happened at the same time in all the countries of the world? Can you specify the time to provide more clarity to the readers?

Response: Again, thank you for noting this. We felt it was implicit that the dates covered by the study in Table 1 could be utilized for this. To provide additional clarity, we have added an additional column to Table 1 with the date of the first case in each country. The limitation of this is that it is not possible to succinctly report this for multi-country studies.

Line 387 to 388: Warnings have been given that populations would not be able to sustain transmission prevention behaviours, such as reduced mobility.

Can you provide a reference for this?

Response: We have added a reference to the following World Health Organisation document to support this statement.

Line 390 to 392: A cross-sectional study of adults in Australia, USA and UK showed that although adherence was restrictions remained high into the second wave, it did decline, particularly in young people and males. Correct the typo in line 391.

Response: Thank you for bringing this to our attention. We have now amended the typo.

Line 491: Summan A, Nandi A. Timing of non-pharmaceutical interventions to mitigate COVID-19 transmission and their effects on mobility: A cross-country analysis. medRxiv 2020.05.09.2009642. 

Please change the DOI of the reference.

Response: We have not provided a doi for this reference. The numbers are part of medRxiv referencing system. We have added the doi for clarity.

Reviewer 2 

The manuscript by Tully et al. “The effect of different COVID-19 public health restrictions on mobility: a systematic review” is well-organized and presented. This manuscript can be accepted in PloS One after minor revision as follows: 

Response: Thank you for reviewing our article and providing a commentary.

Lines 60-70, please elaborate what kinds of various measures? In addition to testing and quarantining the uses of some traditional medicine -based biomolecules as anti-covid 19 agents were founded effective and few drugs also in absence of vaccine i.e. doi: 10.1007/s12088-020-00893-4. In addition, diet and gut microbiota also played important role in combating such infections i.e. doi: 10.1007/s12088-020-00908-0. Please add minor summary of such information’s. 

Response: We appreciate that there have been a number of other approaches in individual countries, but the tone of this paragraph was to give a generalized summary of common measures across the globe. To that end we have therefore added some clarity to the paragraph. We therefore believe adding information, such as that suggested, detracts from the purpose of the paragraph and does not align to the aim of the paper. The aim of this paper is to review the impacts of measures to restrict transmission on google mobility.

Please highlight the clear objective and significance of this article at the end of Introduction.

Response: As requested, we have highlighted the potential significance of this review to the end of the introduction. However, we currently have a stated aim at the end of the introduction and we do not believe adding an objective would add anything further to this paragraph.

Tables may be reorganized more specifically in the term of text contents.

Response: Thank you for your comment. We have organised all tables to allow the reader to reflect the characteristics and comparisons possible between included studies. In response, we have edited the text/layout of the tables and trust this has improved their clarity.

---

## [Decision Letter · Decision Letter 1]

22 Nov 2021

The effect of different COVID-19 public health restrictions on mobility: a systematic review

PONE-D-21-23011R1

Dear Dr. Tully,

We’re pleased to inform you that your manuscript has been judged scientifically suitable for publication and will be formally accepted for publication once it meets all outstanding technical requirements.

Kind regards,

Sanjay Kumar Singh Patel, Ph.D.

Academic Editor

PLOS ONE